# Kupffer Cells Sense Free Fatty Acids and Regulate Hepatic Lipid Metabolism in High-Fat Diet and Inflammation

**DOI:** 10.3390/cells9102258

**Published:** 2020-10-08

**Authors:** Kira L. Diehl, Julia Vorac, Kristina Hofmann, Philippa Meiser, Iris Unterweger, Lars Kuerschner, Heike Weighardt, Irmgard Förster, Christoph Thiele

**Affiliations:** Life and Medical Science Institute (LIMES), University of Bonn, Carl-Troll-Straße 31, 53115 Bonn, Germany; kira.diehl@freenet.de (K.L.D.); juliavorac@googlemail.com (J.V.); kristinahofmann5@gmail.com (K.H.); philippa.meiser@tum.de (P.M.); iris.unterweger@sund.ku.dk (I.U.); kuerschn@uni-bonn.de (L.K.); hwei@uni-bonn.de (H.W.); irmgard.foerster@uni-bonn.de (I.F.)

**Keywords:** non-alcoholic fatty liver disease, nutrition/lipids, metabolic disease, lipids/chemistry

## Abstract

A high fat Western-style diet leads to hepatic steatosis that can progress to steatohepatitis and ultimately cirrhosis or liver cancer. The mechanism that leads to the development of steatosis upon nutritional overload is complex and only partially understood. Using click chemistry-based metabolic tracing and microscopy, we study the interaction between Kupffer cells and hepatocytes ex vivo. In the early phase of steatosis, hepatocytes alone do not display significant deviations in fatty acid metabolism. However, in co-cultures or supernatant transfer experiments, we show that tumor necrosis factor (TNF) secretion by Kupffer cells is necessary and sufficient to induce steatosis in hepatocytes, independent of the challenge of hepatocytes with elevated fatty acid levels. We further show that free fatty acid (FFA) or lipopolysaccharide are both able to trigger release of TNF from Kupffer cells. We conclude that Kupffer cells act as the primary sensor for both FFA overload and bacterial lipopolysaccharide, integrate these signals and transmit the information to the hepatocyte via TNF secretion. Hepatocytes react by alteration in lipid metabolism prominently leading to the accumulation of triacylglycerols (TAGs) in lipid droplets, a hallmark of steatosis.

## 1. Introduction

Non-alcoholic fatty liver disease (NAFLD) is the most common liver disease in industrialized countries [1,2,3] presenting with a wide range of pathologies. This includes steatosis, which manifests in an excessive accumulation of lipid droplets (LDs) in hepatocytes, steatohepatitis, which is a combination of steatosis and liver inflammation, and liver cirrhosis, i.e., excessive fibrosis with loss of hepatocytes and liver function.

The “two-hit model“ was postulated many years ago to describe NAFLD pathogenesis [4]. The “first hit” is characterized by a reversible steatosis, thought to originate from excess amounts of free fatty acid (FFA), which are esterified to triacylglycerol (TAG). This links NAFLD to obesity, in which FFA are strongly elevated [5,6]. In the “second hit”, steatosis can progress into non-alcoholic steatohepatitis, which additionally includes inflammation, scar formation, oxidative stress and hepatic injury, ultimately leading to cirrhosis and liver cancer [4,7,8]. The “second hit” involves the activation of immune cells, in particular liver macrophages, which consist of blood-derived monocytes and liver-resident Kupffer cells [9,10]. The latter respond to extrahepatic inflammation or metabolic stress by coordinating a local immune response via the secretion of various soluble factors, most notably tumor necrosis factor (TNF) [11]. Contradicting the two-hit model, it was recently shown that interleukin(IL)-6 or TNF, originating from adipose tissue, can trigger liver inflammation in parallel with or even before hepatic steatosis [12].

A potent inducer of inflammation is lipopolysaccharide (LPS)—a main component of the outer membrane of Gram-negative bacteria [13,14]. Excess consumption of fat and sugars leads to increased levels of gut-derived LPS in the circulation of obese and NAFLD patients [15].

There are three known causes for elevated serum LPS levels originating from the gut microbiota: Firstly, a Western style diet stimulates bacterial overgrowth and alters the microbiome in the gut. Secondly, sugars such as fructose trigger a disruption of the tight junctions in the gut, which leads to an increased intestinal permeability (“leaky gut”). Thirdly, LPS is absorbed during fat digestion alongside chylomicrons, subsequently reaches the bloodstream [16], enters the liver and activates Toll-like receptor 4 (TLR4) signaling in Kupffer cells [10,17]. In turn, this activation results in the generation of cytotoxic factors such as reactive oxygen species, various inflammatory cytokines and chemokines that induce acute inflammation in the liver [18,19].

Recent studies reported that a selective depletion of Kupffer cells reduces the incidence of steatosis, liver injury, monocyte infiltration and even insulin resistance [11,20,21]. In particular, Tosello-Trampont et al. observed an important role for TNF in generating steatohepatitis in mice on a methionine/choline-deficient diet [11]. Furthermore, Jia et al. were able to show that mice deficient in hepatocyte TLR4 are protected from high-fat diet (HFD)-induced inflammation and insulin resistance [22].

In this paper, we aim to elucidate the individual contributions and the cellular interactions between hepatocytes and Kupffer cells driving the early steatotic phenotype. For this purpose, we use two different but related strategies. The first strategy is a mild chronic steatosis developed in mice after 10 weeks of HFD, the second one is the acute steatosis after in vivo LPS injection. In order to discriminate between the contributions of the individual cell types, we use freshly isolated hepatocytes or Kupffer cells, or co-cultures of both cell types, and supernatant transfer experiments between Kupffer cells and hepatocytes.

The data demonstrate that Kupffer cell activation by excess free fatty acids or LPS is necessary and sufficient to induce hepatocellular steatosis by TNF signaling.

## 2. Materials and Methods

### 2.1. Mouse Experiments In Vivo

Male wild-type C57BL/6NCrl mice were purchased from Jackson Laboratory (Charles River, Erkrath, Germany), maintained under pathogen-free conditions in the animal facility of the LIMES Institute and fed with a breeding and maintenance diet (LASQCdiet^®^ Rod16, LASvendi, Soest, Germany) ad libitum. Animals were used for experiments at the age of 8 weeks. For feeding experiments, 6-week-old mice were fed a controlled normal diet (C1090-10, 10 kcal% fat, ND) for 2 weeks and either a ND or high-fat diet (C1090-60, 60 kcal% fat, HFD, both Altromin, Lage, Germany) for the subsequent 10 weeks ad libitum. Food intake was measured by determining the weight of the remaining food pellets after one week of feeding and kcal were calculated using the respective values of energy density. Animal experiments were performed in conformity with PHS policy and in accordance with the national laws. All protocols were approved by the Institutional Animal Care and Use Committee (Permission LANUV NRW, 84-02.04.2015.A381. The date of approval is 17 March 2016.).

#### 2.1.1. Glucose Tolerance Tests

For glucose tolerance tests, ND and HFD mice were fasted overnight for 12 h. Mice were injected intraperitoneally with 2 mg glucose per g body weight. Tail blood was collected and glucose levels measured using a glucometer (Roche Diagnostics, Mannheim, Germany).

#### 2.1.2. LPS and TNF In Vivo

Mice were injected intraperitoneally with 15 mg/kg *Escherichia coli* LPS serotype 011-B4 (Sigma-Aldrich, St. Louis, USA) 1.5 h before liver perfusion.

#### 2.1.3. Isolation and Culture of Hepatocytes and Kupffer Cells

Hepatocytes and Kupffer cells were isolated by a two-step collagenase liver perfusion as described [23,24] with minor modifications. Briefly, after injecting heparin-sodium (30 Units/g body weight, Ratiopharm, Ulm, Germany), mice were anesthetized with ketamin/xylazin. A butterfly cannula (0.3 mm × 13 mm) was pricked into the portal vein and perfusion was started by pumping Hank’s balanced salt solution with 5 mM EGTA, pH 7.4 and 1 µL/mL heparin-sodium for 5 min at 4 mL/min using a peristaltic pump, followed by perfusion with collagenase buffer (William’s Medium E, 0.125 mM CaCl_2_, 0.5 mg/mL collagenase NB46, Serva Electrophoresis, Heidelberg, Germany, cat no. 17465) for 10 min. Afterwards, cells were released into 50 mL suspension medium (William’s Medium E supplemented with 10% (*v/v*) fetal calf serum, 2 mM L-Glutamine, 100 Units/mL penicillin, 100 µg/mL streptomycin). The cell suspension was centrifuged at 14× *g* for 5 min. The pellet, containing the hepatocytes, was re-suspended in fresh suspension buffer and the suspension was filtered through a 100-µm Nylon cell strainer (Becton Dickinson, New Jersey, USA). The supernatant of the first centrifugation, containing the Kupffer cells, was re-centrifuged at 600× *g* for 10 min. The resulting pellet was re-suspended in Tris-buffered ammonium chloride lysis buffer (17 mM Tris, 160 mM NH_4_Cl, pH 7.2, for erythrocyte lysis) and incubated at room temperature for 5 min prior to the addition of 10 mL PBS and subsequent centrifugation at 600× *g* for 10 min. CD11b MicroBeads (Miltenyi Biotech, Bergisch Gladbach, Germany, order no.: 130-049601) were added to the pellet and Kupffer cells were purified according to the manufacturer’s protocol. Viability of hepatocytes and Kupffer cells was around 80–90% as assessed by trypan blue dye exclusion. Kupffer cells and hepatocytes were plated on collagen-coated dishes, maintained in suspension medium and kept at 37 °C and 5% (*v/v*) CO_2_. Two hours after plating, media of both cell types were renewed to discard non-adherent cells. Purity of hepatocytes and Kupffer cells was checked by flow cytometry (Appendix A) (FACS Canto II, Becton Dickinson, New Jersey, USA).

#### 2.1.4. Serum Parameters

After intraperitoneal injection of heparin-sodium, mouse blood was collected by cardiac puncture under anesthesia as previously described [25]. Serum cholesterol, alanine aminotransferase and aspartate aminotransferase levels were measured using the Reflotron^®^ Plus System (Roche Diagnostics, Rotkreuz, Switzerland). Measurement of serum FFA was performed using the FFA quantitation kit (Sigma-Aldrich, St. Louis, USA).

#### 2.1.5. Determination of Total Hepatic Lipid Content

Livers were immediately homogenized in 4 mL methanol/chloroform (5:1, *v/v*) and phase separation was induced by addition of 3.2 mL water and 0.8 mL chloroform. The chloroform phase was collected and dried in a vacuum centrifuge. The weight of the resulting lipid pellet was determined.

### 2.2. In Vitro Experiments 

#### 2.2.1. LPS and TNF In Vitro

Primary hepatocytes and/or Kupffer cells were incubated in the presence or absence of 100 ng/mL LPS, or 10–50 ng/mL recombinant murine TNF (ImmunoTools, Friesoythe, Germany) and/or 100 ng/mL recombinant rabbit anti-murine TNF antibody (Cat.#500-P64, PeproTech, Hamburg, Germany) at 37 °C for 16 h.

#### 2.2.2. Hepatocyte/Kupffer Cell Co-Culture and Supernatant Transfer Experiments

For co-culture, 2.5 × 10^5^ hepatocytes were cultured with 7 × 10^4^ Kupffer cells in 6-well plates as described [26,27] with minor modifications. For supernatant transfer experiments, a different ratio of cell numbers was used, as described previously [28]. A total of 1 × 10^6^ Kupffer cells per well were cultured in 6-well plates for 16 h. The Kupffer cell supernatant was harvested and, after pelleting remnant cells at 600× *g* for 5 min, transferred to hepatocytes (2.5 × 10^5^) in 6-well plates and incubated at 37 °C for 16 h.

#### 2.2.3. Flow Cytometry

Hepatocytes were seeded in 6-cm dishes at a density of 3.5 × 10^5^ cells/well. After incubation at 37 °C for 12 h, the cells were detached by scraping and pelleted by centrifugation at 11× *g* for 5 min. Kupffer cells were seeded in 12-well plates at a density of 3.5 × 10^5^ cells/well. After incubation at 37 °C for 12 h, the cells were detached by scraping and pelleted by centrifugation at 600× *g* for 10 min. Cells were re-suspended in 200 µL PBS. Anti-mouse CD45RB-PE (BD Biosciences, Franklin Lakes, USA) and CD95-APC antibodies or CD11b-PE and F4/80-APC (Ebioscience, San Diego, USA) were added according to the manufacturer’s instructions. After incubation at 4 °C for 20 min, cell fluorescence was evaluated by flow cytometry using FACS Canto II (BD Biosciences, Franklin Lakes, USA).

#### 2.2.4. Pulse-Chase Experiments with Alkyne Lipids

Hepatocytes (3.5 × 10^5^) were cultured in 6-cm dishes for 2 h and pulse-labeled for 2 min using 66 µM alkyne-oleate and 33 µM alkyne-palmitate [29] in William’s Medium E containing 2 mM L-Glutamine, 100 Units/mL penicillin, 100 µg/mL streptomycin and 1% delipidated BSA. Pulse media were removed, cells were washed with PBS + 1% delipidated BSA and chased in chase medium (66 µM oleate, 33 µM palmitate in William’s Medium E containing 2 mM L-Glutamine, 100 Units/mL penicillin, 100 µg/mL streptomycin, 1% fatty acid-free BSA) for different periods as indicated. Chase media were removed and cells were washed twice with PBS. Lipids were extracted, subjected to fluorogenic click reaction according to the standard procedure [29]. Briefly, the dry lipid pellet was redissolved in 7 µL chloroform, followed by addition of 30 µL click reaction mixture (5 µL of 44.5 mM 3-azido-7-hydroxycoumarin, 500 µL of 10 mM [acetonitrile]_4_CuBF_4_ in acetonitrile, 2 mL ethanol). The tube was incubated at 42 °C for 3 h and the reaction mix applied onto a 20 × 20 cm silica thin layer chromatography (TLC) plate (no UV-indicator, Merck No. 1.05721.0001). The plate was developed in CHCl_3_/MeOH/water/acetic acid 65/25/4/1 for 10 cm, dried for 2 min in a warm stream of air, and developed again for 18 cm in hexane/ethyl acetate 1/1. The plate was briefly dried in a stream of warm air and soaked for 5 sec in 4% (*v/v*) *N,N*-diisopropylethylamine in hexane. The plate was placed in a hood for 1 min to evaporate excess solvent, followed by fluorescent imaging (exc. 420 nm, em. 482–502 nm. Bands were identified by co-migrating synthetic standards. VLDL secretion was determined by analyzing 120 min chase media for alkyne-labeled TG. For determination of fatty acid uptake, pulse times were extended to 10 min.

#### 2.2.5. ELISA

In vivo (1.5 h) or in vitro (16 h) LPS-treated Kupffer cells (7 × 10^4^) were cultured in 96-well plates for 16 h and the supernatants examined for cytokine secretion using commercial enzyme immunoassay kits (R&D Systems, Minneapolis, USA).

#### 2.2.6. Fluorescence Microscopy

Hepatocytes (2.5 × 10^5^ per well of a 6-well dish) were plated with or without 7 × 10^4^ Kupffer cells, on collagen-coated glass coverslips. Cells were fixed in 4% PFA in PBS for 1 h, followed by staining with 0.1 µg/mL LD540 [30] and 1 µg/mL 4′,6-diamidino-2-phenylindole (DAPI). Coverslips were rinsed in water and mounted in fluorescent mounting medium (Dako, Hamburg, Germany). Epifluorescence images were acquired on an Axio Observer Z1 (Zeiss, Oberkochen, Germany) in Apotome mode with a 63×, NA 1.4 Plan-Apochromat objective and a Hamamatsu Orca flash 2.0 camera. Quantification of lipid droplet size, number and total LD540 fluorescent area was performed using Fiji [31]. If indicated, data were normalized to total number of cells using equal threshold settings for all images in one figure.

### 2.3. Statistical Analysis

Statistical analysis was performed using GraphPad Prism software (version 5.0) (GraphPad, La Jolla, USA). All data are presented as mean values with SEM. Significance was analyzed with a two-tailed Student’s *t*-test. Statistical significance is indicated as follows: ns, not significant, * *p* < 0.05, ** *p* < 0.01, *** *p* < 0.001.

## 3. Results

### 3.1. High-Fat Diet (HFD) Mice Show Mild Steatotic Characteristics

In our HFD model, mice were fed a 60% high-fat diet for 10 weeks. During the experiment, the mice consumed normal amounts of food with elevated caloric density (Appendix A). This resulted in a moderate weight gain of about 6 g (Appendix A) along with a rather mild steatosis, characterized by unchanged liver weight (HFD: 1.65 ± 0.09 g vs. normal diet (ND): 1.50 ± 0.04 g *n* = 4), normal liver appearance (Appendix A) and a 1.8-fold increase in hepatic lipid content (Figure 1A). Serum FFA and cholesterol were elevated (Figure 1B), intraperitoneal glucose tolerance was decreased (Figure 1C). There was a significant increase in alanine aminotransferase and a slight increase in aspartate aminotransferase (Figure 1D). Upon isolation of hepatic cells, numbers of both hepatocytes and CD11b-positive Kupffer cells were unchanged (Figure 1E). When stained with the LD-specific dye LD540, isolated hepatocytes of HFD animals showed lipid droplets (LDs) with increased size and total area (Figure 1F).

In order to characterize the fatty acid metabolism on a cellular scale, we performed a pulse-chase experiment of fatty acid incorporation into the major lipid classes along the phospholipid and TAG biosynthetic pathway, using alkyne-labeled fatty acids and fluorescent detection technology [29] (Figure 2A). For labeling, a 1/2 mixture of alkyne-palmitate and alkyne oleate was used, which ensures with their typical preferences for sn-1 and sn-2 positions [29], respectively, an efficient labeling of most phospholipids. Labeling in TAG is expected to distribute over all three hydroxyl groups. Under the conditions of a short labeling pulse, multiple labeling in one molecule was not observed. Note that the labeling pattern indicates the relative flow of label into the target molecules. For determination of absolute lipid synthesis rates, additional information regarding total lipid synthesis rate would be required. Hepatocytes of ND mice showed a rapid initial incorporation of labeled fatty acids mostly into diacylglycerol (DAG), phosphatidylcholine (PC) and TAG. Upon a chase, the labeled DAG was converted mainly to PC and TAG, which are the major final products of exogenous fatty acids. Hepatocytes of HFD mice showed similar results. However, notable differences were found in DAG metabolism: the initial amount of labeling in DAG was smaller (Figure 2B) and the ratio of 1,3-DAG to 1,2-DAG was larger in HFD hepatocytes than in control hepatocytes (Figure 2A). While 1,2-DAG is an anabolic biosynthetic DAG derived from phosphatidic acid (PA) by dephosphorylation, 1,3-DAG is likely a product of catabolic action of the adipose TAG lipase on TAG [32] (Figure 2C). Therefore, the changed ratio of these DAGs indicates an increased TAG turnover in HFD hepatocytes. Metabolism of phosphatidic acid and phosphatidylethanolamine (PE), as well as final amounts of labeled PC and TAG were very similar under the different diets (Figure 2B). Overall, the pulse chase experiment with isolated hepatocytes failed to explain the steatotic phenotype of HFD mice.

### 3.2. TNF Is a Major Player in Generating a Steatotic Phenotype in ND Hepatocytes

A possible explanation for this failure is a major contribution of other cell types in the liver, notably Kupffer cells, which communicate with hepatocytes via secreted factors. Therefore, we cultivated ex vivo Kupffer cells from ND and HFD mice and analyzed factors secreted into the culture medium. There were no differences in the secretion of IL-6, IL-10 and IL-1β, but an elevated secretion of TNF by HFD Kupffer cells compared to Kupffer cells from ND animals was found (Figure 3A). To determine whether this release of TNF was sufficient to explain the steatosis observed in HFD hepatocytes, we incubated hepatocytes from ND mice with culture supernatants of Kupffer cells either from ND or HFD mice (Figure 3B). Only the HFD Kupffer cell supernatant induced an increase in LD size and total area per cell (Figure 3B, frame 3), which was reverted upon co-addition of a neutralizing anti-TNF antibody (frame 4). Image quantification confirmed these observations regarding total LD area, LD diameter and LD number (Figure 3C). We conclude that Kupffer cell-derived TNF is necessary to mediate steatosis in hepatocytes that have not been challenged with a HFD.

To determine whether TNF alone is sufficient to induce steatosis in hepatocytes from ND mice, we treated isolated hepatocytes with various concentrations of TNF, resulting in a dose-dependent increase in LD size and area (Figure 4A,B). This finding indicates that Kupffer cell-derived TNF, rather than a metabolic overload of the hepatocytes, is the key factor in the development of steatosis. This raises the question of how fatty acid overload is able to activate Kupffer cells leading to increased TNF. Treatment of isolated Kupffer cells with 0.5 mM of a 2/1 mixture of oleic and palmitic acid, resembling plasma concentrations under HFD (Figure 1B), resulted in a significant increase in released TNF (Figure 4C), indicating that Kupffer cells act as hepatic fatty acid sensors.

### 3.3. Short Term LPS Stimulation In Vivo Alters the Hepatic Lipid Metabolism

Besides HFD, also inflammatory stimuli including bacterial surface antigens such as LPS can lead to TNF secretion in the liver. LPS release into the portal vein blood is frequently discussed as a factor contributing to steatohepatitis caused by metabolic overload [33,34]. Therefore, we injected ND and HFD mice with 15 mg/kg LPS and analyzed total hepatic lipid weight and serum parameters (Figure 1A,B,D). Both ND and HFD animals showed an increase in liver total lipids and serum FFA, unchanged serum cholesterol and increased markers of liver damage upon LPS treatment. To compare these phenomena with the findings on ND vs. HFD mice described above, we studied isolated hepatocytes and Kupffer cells of untreated (control) vs. LPS-injected mice. LPS injection led to a significant increase in stored TAG (Figure 5A–C). It should be noted that the absolute values are not identical to those in Figure 1A, because animals used in Figure 5 had not been treated with ND for an additional 10 weeks as in Figure 1. These younger animals had less basal liver TAG (see also Appendix A). Thus, lipid accumulation is age-dependent, as was previously observed [35,36].

By metabolic tracing of alkyne-labeled FFA, we observed an increased fatty acid incorporation (Figure 5D) and a significantly decreased TAG secretion (Figure 5E) in LPS-treated animals, consistent with a net increase in stored TAG. Detailed pulse-chase analysis of fatty acid metabolism in these hepatocytes revealed a marked increase in TAG synthesis and corresponding decrease in PC synthesis after in vivo LPS treatment (Figure 6A,B compare control vs. in vivo LPS). However, in vitro LPS stimulation of isolated hepatocytes by adding LPS into the culture medium for 1.5 h was not able to reproduce these observations and resulted in a normal fatty acid metabolism (Figure 6, in vitro LPS) and a lack of steatosis upon microscopic analysis of TAG storage in LDs (Figure 7A,B, frame 1+2). In contrast, upon LPS treatment of co-cultures of hepatocytes with Kupffer cells, we observed increased TAG storage in LDs (Figure 7A, frame 3+4), due to an increased size of droplets (Figure 7B, middle panel), and marginally increased LD number (right panel).

Kupffer cell culture supernatants isolated from LPS-treated mice (Figure 3A) showed significantly elevated levels of TNF accompanied by slightly elevated IL-1β, IL-6 and IL-10 levels. Likewise, supernatants of in vitro LPS-treated Kupffer cells (Figure 7C) showed significantly elevated levels of TNF and elevated amounts of IL-10 and IL-1β. We repeated the experiment shown in Figure 3B, this time using LPS-stimulation instead of HFD treatment (Appendix A). Supernatant of LPS-stimulated Kupffer cells induced LD accumulation in hepatocytes of untreated mice, which was prevented by co-addition of a neutralizing TNF antibody.

## 4. Discussion

The present study aimed at defining the factors that trigger the development of initial pathological signs in the liver during metabolic overload. During HFD treatment, and along with the initiation of macroscopic obesity, mice showed hepatic symptoms of a steatosis without major signs of inflammatory steatohepatitis. We observed marginally elevated serum transaminases, macroscopically normal size and appearance of the organ, combined with slightly increased total liver lipids and with a clearly elevated number of cytoplasmic LDs in hepatocytes. In contrast to our expectations, detailed biochemical experiments in hepatocytes of these organs showed a quantitatively and qualitatively normal fatty acid metabolism. Increased secretion of TNF by Kupffer cells was the only marker that pointed to an inflammatory component of the pathology at this early stage. Strikingly, Kupffer cell culture supernatant was sufficient to mirror the HFD phenotype in healthy hepatocytes and TNF was identified as the crucial component to mediate this effect.

A major advantage of cultivated primary cells is to discriminate between hepatocellular metabolic overload stress and Kupffer cell-derived signals as causative agents for hepatocyte steatosis. The results clearly indicate that at least in this early phase of obesity, the contribution of Kupffer cells is more important than the hepatocyte overload stress. Our data do not exclude steatotic contributions of cells outside the liver, in particular factors released from adipose tissue, however, it appears very likely that Kupffer cells are the major origin of signals that promote hepatic steatosis upon a high-fat diet, consistent with in vivo studies on development of steatohepatitis upon choline/methionine-free diet [11].

In vivo LPS treatment leads to hepatic steatosis that is phenotypically similar to the one observed after 10 weeks of HFD. Kupffer cells are essential for the development of steatosis, since LPS did not lead to steatosis when applied to pure hepatocytes. However, the profile of cytokines released by Kupffer cells shows a broader activation of the inflammatory response, and serum transaminases are more strongly elevated than upon HFD treatment. On the biochemical level, the LPS-induced steatosis is reflected by an altered fatty acid pulse-chase profile, which was not observed in HFD hepatocytes. Despite these differences, our data indicate that the LPS-induced steatosis also depends on Kupffer cell-derived TNF. The comparison between HFD and LPS treatment suggests that TNF, regardless of the cause of its release, is the major mediator of hepatic steatosis.

In this context, it is intriguing that fasting mice develop an intermittent steatosis with increased FFA [37] and have strongly elevated serum TNF concentrations [38]. As shown in Figure 4C, treatment of isolated Kupffer cells with 0.5 mM FFA, a concentration found in serum upon challenge with HFD or LPS or both (Figure 1B), or in starving mice [37] resulted in increased secretion of TNF. This is consistent with previous experiments on induction of *TNF* mRNA expression in Kupffer cells by FFA [39]. Furthermore, FFAs were shown to induce TLR4 signaling in RAW264.7 cells [40] and the expression of TLR4 in Kupffer cells [39]. In addition, HFD-fed mice are more susceptible to a sustained low-dose LPS-induced increase in TNF [41], which appears to be mediated by leptin-induced expression of CD14, a co-receptor of TLR4. Conclusively, elevated FFA, originating from nutritional overload or released from adipose tissue during fasting, would be the primary cause of TNF release from Kupffer cells that leads to hepatocyte steatosis. TLR4 is a likely candidate as an FFA receptor on Kupffer cells to sense fatty acid abundance and overload in the liver. As illustrated in Figure 8, Kupffer cells appear to act as hepatic FFA sensory cells that monitor the fatty acid load and trigger hepatocyte response.

## Figures and Tables

**Figure 1 cells-09-02258-f001:**
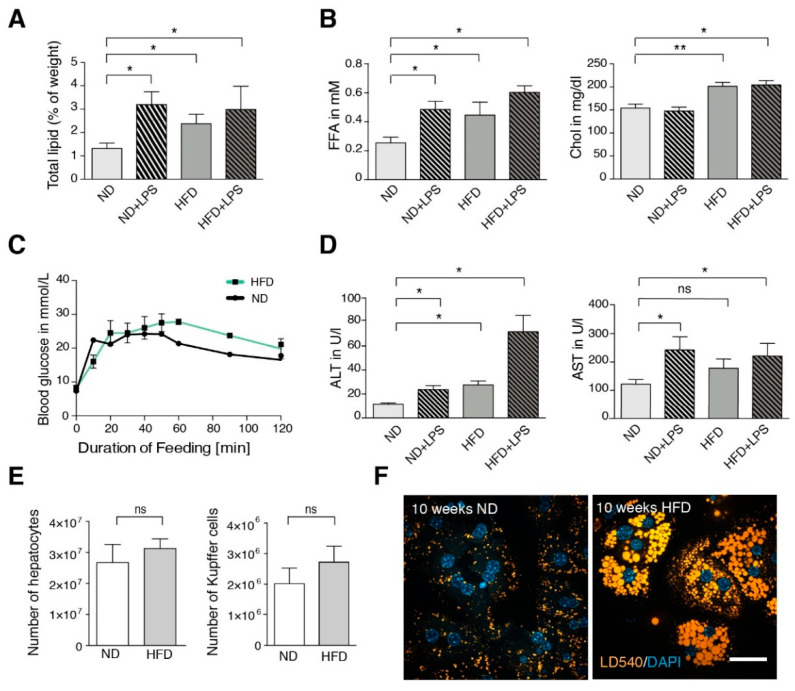
Characterization of liver and blood parameters in normal diet (ND) and high-fat diet (HFD) mice. (**A**) Averaged total lipid (weight %) in livers of ND and HFD mice (*n* ≥ 3). (**B**) Serum levels of free fatty acid (FFA) and cholesterol (Chol) of ND and HFD mice (*n* ≥ 3). (**C**) Intraperitoneal glucose tolerance test for ND and HFD mice (*n* = 3). (**D**) Serum levels of alanine aminotransferase (ALT) and aspartate aminotransferase (AST) in ND and HFD mice (*n* ≥ 3). (**E**) Total number of hepatocytes and Kupffer cells isolated per liver (*n* ≥ 3). (**F**) Lipid droplet staining (LD540) in hepatocytes after 10 weeks of feeding. If indicated (+lipopolysaccharide (+LPS)), mice were injected intraperitoneally with 15 mg/kg weight of E. coli LPS 1.5 h before liver perfusion. Scale bar: 50 µm. Data are represented as mean with SEM of at least three independent experiments as indicated. Significance was analyzed with a two-tailed Student’s *t*-test. Statistical significance is indicated as follows: ns, not significant, * *p* < 0.05, ** *p* < 0.01.

**Figure 2 cells-09-02258-f002:**
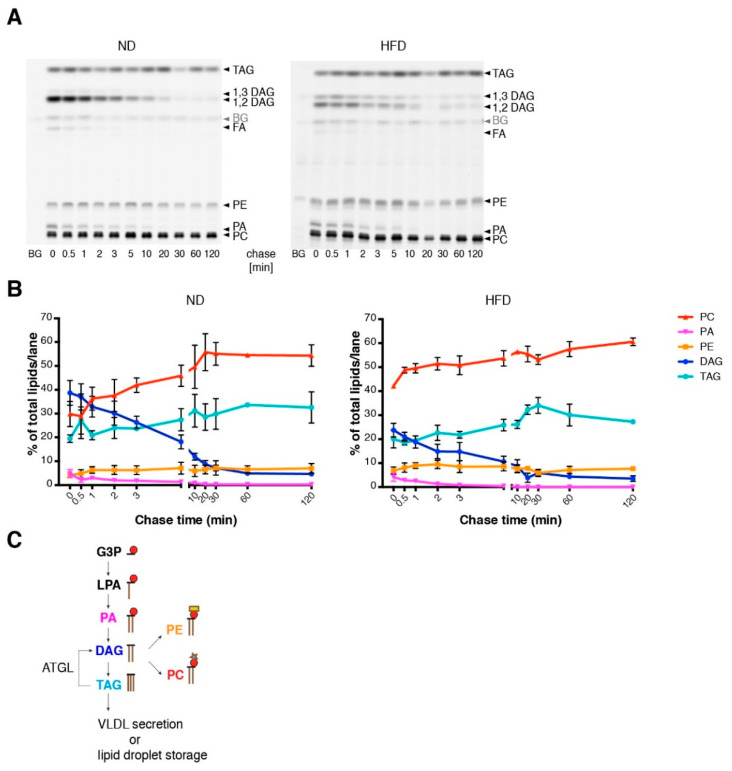
Lipid metabolism in isolated hepatocytes of ND and HFD mice. (**A**) Hepatocytes from ND or HFD mice were cultured for 2 h prior to pulse-chase experiments. After pulse incubation (2 min) with alkyne fatty acids, hepatocytes were washed and subsequently incubated with chase medium for the indicated time. Lipids were extracted and the extracts click-reacted with the dye azidocoumarin. Lipids were separated on a thin layer chromatography (TLC) plate and metabolites of alkyne fatty acids were detected by fluorescence imaging. Metabolites were identified by co-migrating standards. (**B**) Quantification of the TLC plates shown in panel A. Numbers indicate the percentage of the respective lipid class relative to total labeled lipid per lane (*n* ≥ 3). (**C**) Metabolic flow of fatty acids in the liver starting with the incorporation of acyl-CoA into glycerol-3-phosphate (G3P), lysophosphatidic acid (LPA), phosphatidic acid (PA) and ending with either the incorporation into triacylglycerol (TAG) or into the phospholipids phosphatidylcholine (PC) or phosphatidylethanolamine (PE). Lipolytic action of the enzyme ATGL (adipose TAG lipase) leads to the formation of diacylglycerol (DAG) from TAG. Mean values with SEM of triplicate determinations are shown. BG: background.

**Figure 3 cells-09-02258-f003:**
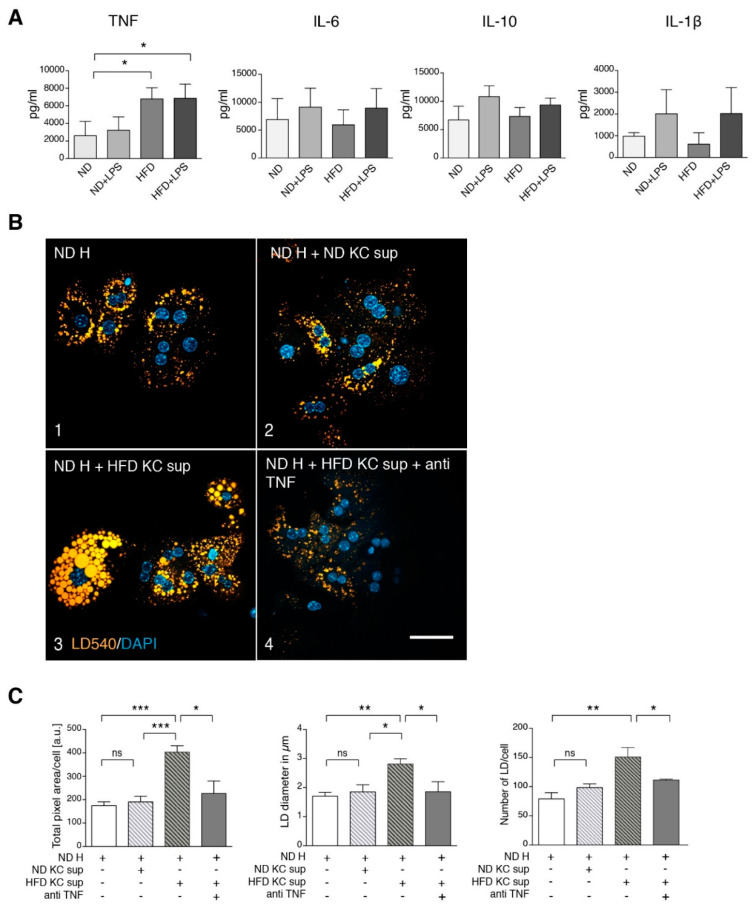
Kupffer cell supernatant from HFD mice stimulates steatosis in ND hepatocytes via tumor necrosis factor (TNF). (**A**) Cytokine secretion by Kupffer cells from untreated and LPS-treated ND and HFD mice. Mice were treated as indicated, Kupffer cells were isolated and cultured for 16 h. Secreted cytokines were measured by ELISA (TNF *n* ≥ 3, IL-1β *n* ≥ 3, IL-6 *n* ≥ 4, IL-10 *n* ≥ 2). (**B**) Freshly isolated hepatocytes (H) from ND mice were cultured in the absence (frame 1) or presence of 16 h pre-conditioned supernatant of Kupffer cells (KC sup) from ND (frame 2) or HFD mice (frame 3 and 4) as indicated. In frame 4, a neutralizing TNF antibody was added. Microscopy images of cells that were stained for lipid droplets (LD540) and nuclei (DAPI) are shown. Scale bar: 50 µm. (**C**) Quantification of the total fluorescent LD area/cell, average LD diameter in µm and LD number/cell of the various conditions (*n* ≥ 3). Mean values with SEM of at least three independent experiments are shown. H: hepatocytes, KC: Kupffer cells, ND: controlled normal diet, HFD: high-fat diet, sup: supernatant. Significance was analyzed with a two-tailed Student’s *t*-test. Statistical significance is indicated as follows: ns, not significant, * *p* < 0.05, ** *p* < 0.01, *** *p* < 0.001.

**Figure 4 cells-09-02258-f004:**
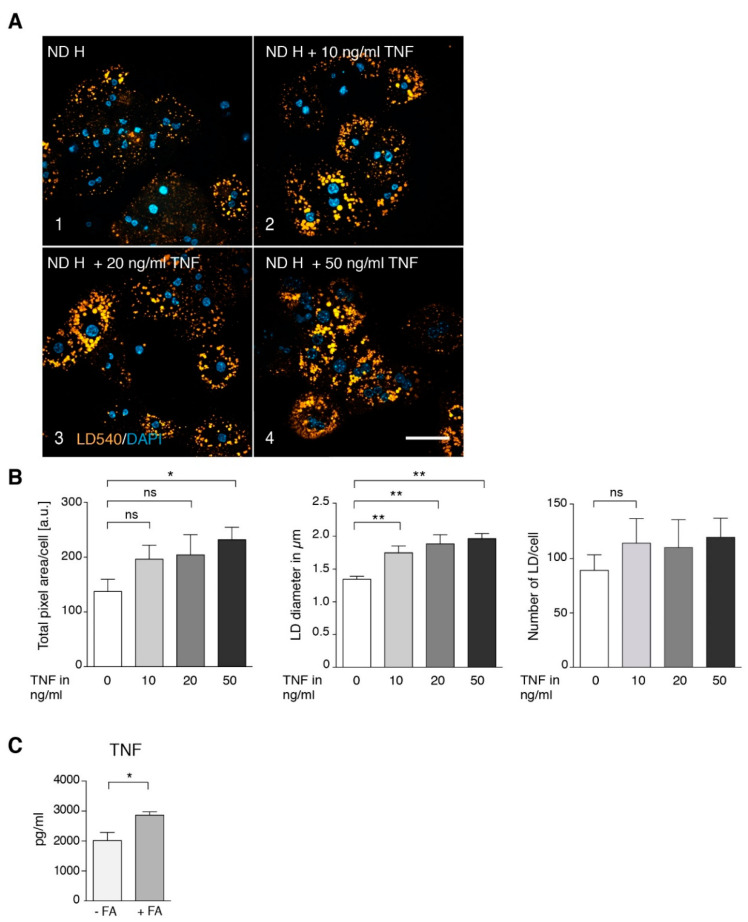
Influence of TNF on lipid droplet size and number in normal diet hepatocytes. Primary hepatocytes from ND mice were isolated and treated with different concentrations of TNF for 2 h. (**A**) Staining for lipid droplets (LD540) and nuclei (DAPI). Scale bar: 50 µm. (**B**) Quantification of total fluorescent lipid droplet (LD) area per cell, LD size in µm and number of lipid droplets/cell (*n* = 6). (**C**) Secretion of TNF by isolated Kupffer cells after 16 h treatment with 0.5 mM FFA. Data are represented as mean value with SEM. Significance was analyzed with a two-tailed Student’s *t*-test. Statistical significance is indicated as follows: ns, not significant, * *p* < 0.05, ** *p* < 0.01.

**Figure 5 cells-09-02258-f005:**
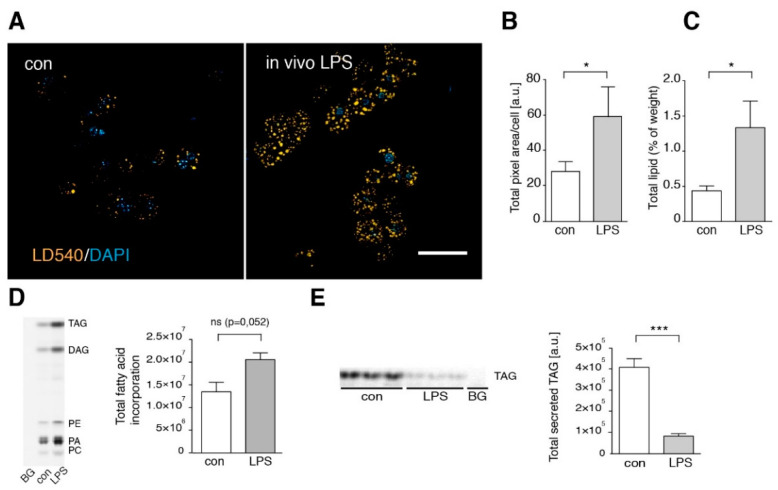
In vivo LPS treatment leads to hepatic steatosis. (**A**) Hepatocytes were isolated from in vivo LPS-treated (LPS) or untreated (control) mice. Cells were stained with LD540 and DAPI to visualize lipid droplet (LD). Scale bar: 50 µM. (**B**) Quantification of the total fluorescent LD area per cell (*n* = 3). (**C**) Averaged total lipids (weight %) in livers of control (con) and LPS-treated animals (LPS) (*n* ≥ 4). (**D**) To determine fatty acid incorporation, hepatocytes were incubated with alkyne fatty acids for 10 min. Lipids were extracted and labeled lipids analyzed as described in Experimental Procedures. Left: image of a representative TLC plate. Right: quantification of total fluorescent lipids/lane (*n* = 3). (**E**) For determination of VLDL secretion, chase medium was collected after 2 min alkyne fatty acid pulse and 120 min chase time, analyzed as above, and the amount of VLDL was determined from the TAG signal. Left: Image of the TLC plate, right: quantification of total secreted TAG (*n* = 3). Data are visualized as mean values with SEM. Significance was analyzed with a two-tailed Student’s *t*-test. Statistical significance is indicated as follows: ns, not significant, * *p* < 0.05, *** *p* < 0.001.

**Figure 6 cells-09-02258-f006:**
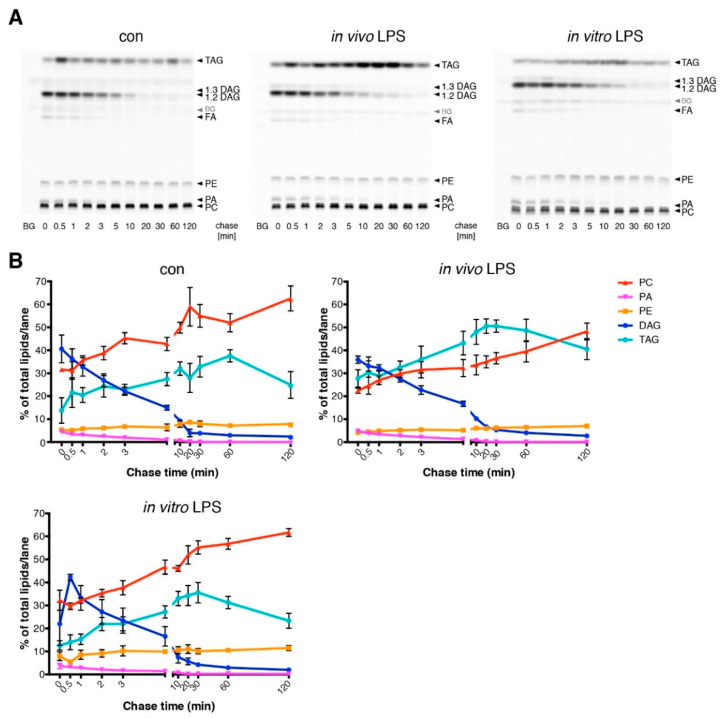
Differential influence of in vivo or in vitro LPS stimulation on TAG synthesis in isolated hepatocytes. (**A**) Pulse-chase experiment with hepatocytes from untreated (con) or 1.5 h in vivo LPS-treated (LPS) mice. For comparison, hepatocytes from untreated animals were treated with 100 ng/mL LPS for 1.5 h (in vitro LPS). Pulse-chase analysis and lipid extraction were performed as described in Figure 2. (**B**) Quantification of the TLC plates shown in panel A (*n* ≥ 3). Data are shown as mean values with SEM.

**Figure 7 cells-09-02258-f007:**
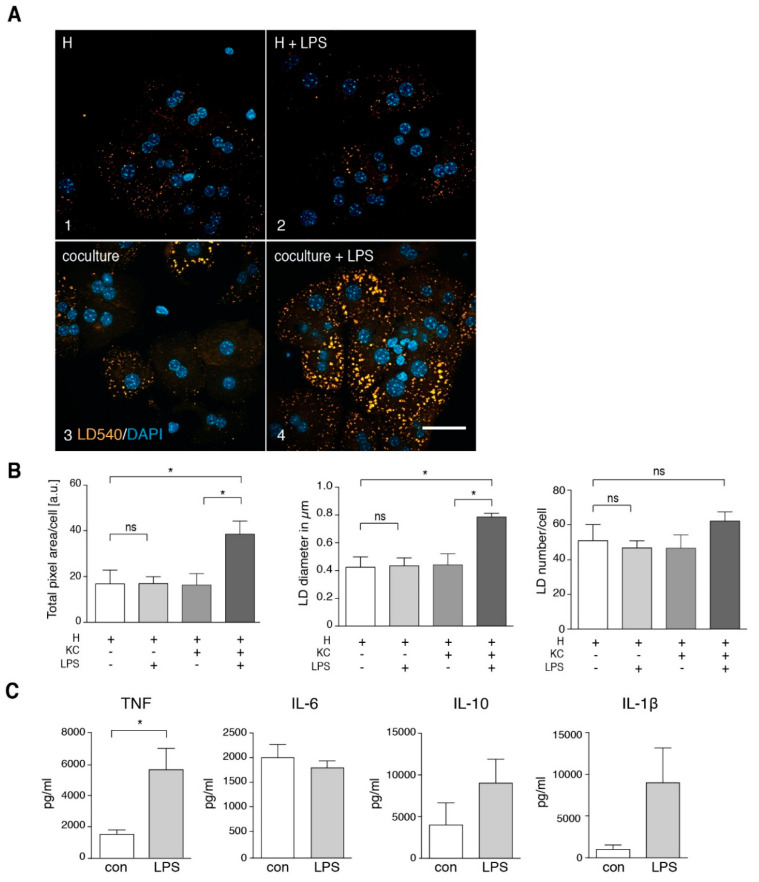
Intracellular lipid droplet accumulation in hepatocytes upon in vitro LPS treatment depends on the presence of Kupffer cells. (**A**) Primary mouse hepatocytes (H) were incubated for 16 h in the absence (frame 1) or presence of LPS in vitro (frame 2), in co-culture with Kupffer cells (KC) (frame 3) or in co-culture with Kupffer cells and LPS in vitro (frame 4). The lipid droplet (LD) accumulation is visualized by staining with LD540 (orange) and DAPI (blue). Scale bar: 50 µm. (**B**) Quantification of total fluorescent LD area und LD number per cell as well as LD size (*n* = 3). (**C**) Kupffer cells were isolated, followed by 16 h of cell culture in the presence of 100 ng/mL LPS. Secreted TNF, IL-6, IL-10 and IL-1β was measured in the Kupffer cell culture supernatant by ELISA. TNF *n* = 4; IL-6 *n* = 6; IL-10 *n* = 3; IL-1β *n* = 3. All data are presented as mean value with SEM. Significance was analyzed with a two-tailed Student’s *t*-test. Statistical significance is indicated as follows: ns, not significant, * *p* < 0.05.

**Figure 8 cells-09-02258-f008:**
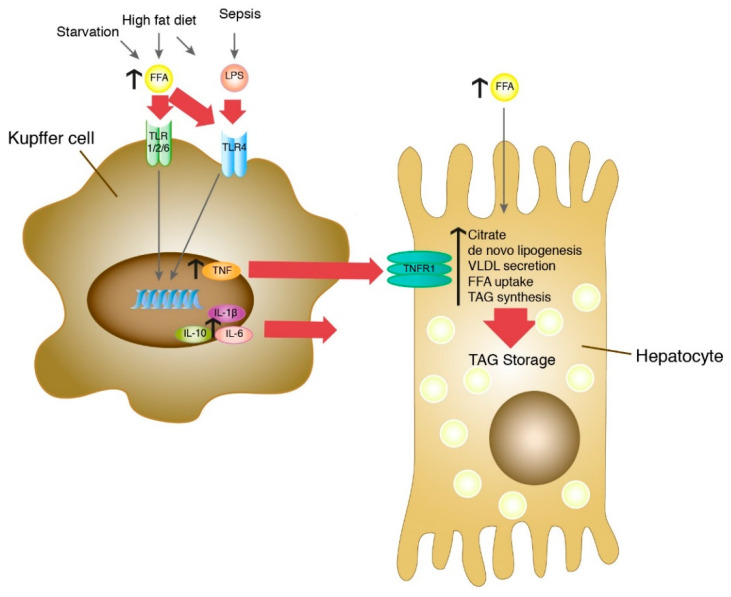
The role of Kupffer cells as hepatic sensory cells in the development of steatosis in hepatocytes. Increased FFA can be caused by lipolysis in adipose tissue during starvation or by excess nutritional intake in high-fat diet. Elevated LPS levels can be caused by high-fat diet or sepsis. Both FFA and LPS bind to Toll-like receptors on the surface of Kupffer cells and lead to increased secretion of IL-1β, IL-10 und IL-6 and TNF. Secreted TNF binds to the TNFR1 receptor on hepatocytes and leads to increased citrate levels, de novo lipogenesis, FFA uptake and TAG synthesis and to a reduced VLDL secretion. As a result, this leads to an increased TAG storage and formation of steatosis in hepatocytes. In this model, the Kupffer cell is the primary sensor for both FFA overload and LPS, integrates these signals and utilizes TNF to transmit the information to the hepatocyte, which reacts by alteration of lipid metabolism.

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
