# Peer review of "Kupffer Cells Sense Free Fatty Acids and Regulate Hepatic Lipid Metabolism in High-Fat Diet and Inflammation"

_cells, 2020, doi:10.3390/cells9102258_

Round 1

Reviewer 1 Report

The manuscript is scientifically sound and the documentation of the experiments and their analysis is very transparent. The authors’ observations are of great interest to other researchers and their conclusions are justified by the results. This is an excellent manuscript.

I have only minor comments:

Line 98: Is it really 14 x g? This seems to be an extremely slow speed for centrifugation, even for isolation of cells.

A list of abbreviations would be helpful to ease the understanding of the manuscript. If no list of abbreviations is envisaged, the authors should make sure that all abbreviations are explained in the manuscript text.

Author Response

The manuscript is scientifically sound and the documentation of the experiments and their analysis is very transparent. The authors’ observations are of great interest to other researchers and their conclusions are justified by the results. This is an excellent manuscript.

I have only minor comments:

Line 98: Is it really 14 x g? This seems to be an extremely slow speed for centrifugation, even for isolation of cells.

Yes, it is 14 x g, corresponding to 350 rpm in our falcon-tube centrifuge

A list of abbreviations would be helpful to ease the understanding of the manuscript. If no list of abbreviations is envisaged, the authors should make sure that all abbreviations are explained in the manuscript text.

We went through the text and amended a couple of missing explanations for abbreviations (in yellow, please see the attachment)

Reviewer 2 Report

This paper presents evidence to support a central role of paracrine secretion of TNF from Kupffer cells in the pathogenesis of hepatic steatosis.  The experimental work appears to have been well-conducted and reasonably well described.  A number of specific points about presentation are detailed below. This evidence is based primarily on microscopic analysis of fluorescence-labelled lipid droplets in hepatocytes, while the click based analyses of lipid metabolism provides useful information but is not central to the primary conclusion of the paper.  In the discussion, for example, the focus is clearly on the TNF actions with little emphasis on fatty acid metabolism.  The authors should consider more closely linking these two experimental approaches, for instance if alkyne fatty acid incorporations can be used to give an estimate of the contribution of increased TAG synthesis to the hepatic steatosis. 

In this context, the incorporation data presented in Figures 2 and 6 is an elegant approach that demonstrates the rapidity of substrate transfer through DAG to TAG and PC, but does not by itself provide an analysis of the rates of lipid synthesis.  Instead, it provides an analysis of the fractional distribution of incorporated label into the different classes, where incorporation into one class may appear to increase as a consequence of decreased incorporation into another. The data presented in Figure 5, for example, is more semi-quantitative demonstrating that the hepatic lipid accumulation after short-term LPS treatment is largely due to impaired VLDL secretion.  It is difficult to determine from these results whether high fat feeding significantly affects the rate of hepatic TAG synthesis either in vivo or in vitro.  One issue the authors should address is the potential effect of dilution of substrate alkyne fatty acid with an increased pool of unlabelled fatty acyl CoA as a consequence of high fat feeding.  This could have the effect of decreased apparent lipid synthesis.

Para 2.8 Presumably these livers were separate from those used for perfusion.  Mashed is inappropriate here should be substituted with homogenised. The meaning of 4 ml/liver methanol/chloroform is not clear.

Para 2.9 Not all readers will be familiar with the fluorogenic click reaction and a brief summary should be provided here in addition to the legend of Figure 2.  The authors should also clarify that the choice of alkyne-oleate and alkyne-palmitate will only label lipids containing the respective fatty acids, which is especially relevant because of the highly unsaturated composition of mouse liver PC and PE. Presumably, the palmitate analogue was chosen to label the sn-1 position of PC and PE with the oleate analogue labelling the sn-2 position.  The authors should also comment that these analogues will only be incorporated into one position of the phospholipids but could label any three positions of the glycerol backbone of TAG.

Figure 1 legend should include details about the LPS treatments and amounts administered.

Figure 3 legend states that hepatocytes were incubated with 16h  pre-conditioned supernatant from Kupffer cells but para 2.5 states that Kupffer cell supernatants were harvested after 2h before transfer to hepatocytes for a 16h incubation.  This apparent contradiction should be clarified.

Author Response

This paper presents evidence to support a central role of paracrine secretion of TNF from Kupffer cells in the pathogenesis of hepatic steatosis.  The experimental work appears to have been well-conducted and reasonably well described.  A number of specific points about presentation are detailed below. This evidence is based primarily on microscopic analysis of fluorescence-labelled lipid droplets in hepatocytes, while the click based analyses of lipid metabolism provides useful information but is not central to the primary conclusion of the paper.  In the discussion, for example, the focus is clearly on the TNF actions with little emphasis on fatty acid metabolism.  The authors should consider more closely linking these two experimental approaches, for instance if alkyne fatty acid incorporations can be used to give an estimate of the contribution of increased TAG synthesis to the hepatic steatosis. 

In this context, the incorporation data presented in Figures 2 and 6 is an elegant approach that demonstrates the rapidity of substrate transfer through DAG to TAG and PC, but does not by itself provide an analysis of the rates of lipid synthesis.  Instead, it provides an analysis of the fractional distribution of incorporated label into the different classes, where incorporation into one class may appear to increase as a consequence of decreased incorporation into another. The data presented in Figure 5, for example, is more semi-quantitative demonstrating that the hepatic lipid accumulation after short-term LPS treatment is largely due to impaired VLDL secretion.  It is difficult to determine from these results whether high fat feeding significantly affects the rate of hepatic TAG synthesis either in vivo or in vitro.  One issue the authors should address is the potential effect of dilution of substrate alkyne fatty acid with an increased pool of unlabelled fatty acyl CoA as a consequence of high fat feeding.  This could have the effect of decreased apparent lipid synthesis.

We agree that the data suggest that reduced VLDL secretion significantly contributes to the steatosis upon LPS treatment. Regarding the apparent lack of increased total incorporation in HFD cells, the reviewer is right that this might be due to a dilution or saturation effect. We added an according comment in the result section: "Note that the labeling pattern indicates the relative flow of label into the target molecules. For determination of absolute lipid synthesis rates, additional information regarding total lipid synthesis rate would be required. We will address this interesting issue with experiments with absolute quantification of all relevant pathways in our future research" (in green, please see the attachment).

Para 2.8 Presumably these livers were separate from those used for perfusion.  Mashed is inappropriate here should be substituted with homogenised. The meaning of 4 ml/liver methanol/chloroform is not clear.

We corrected the text and specified the volumes accordingly (see green changes)

Para 2.9 Not all readers will be familiar with the fluorogenic click reaction and a brief summary should be provided here in addition to the legend of Figure 2. 

We added a compact description of the methodology (see green changes).

The authors should also clarify that the choice of alkyne-oleate and alkyne-palmitate will only label lipids containing the respective fatty acids, which is especially relevant because of the highly unsaturated composition of mouse liver PC and PE. Presumably, the palmitate analogue was chosen to label the sn-1 position of PC and PE with the oleate analogue labelling the sn-2 position.  The authors should also comment that these analogues will only be incorporated into one position of the phospholipids but could label any three positions of the glycerol backbone of TAG.

Thank you for the thoughtful comments. We agree and added  an according explanation into the results section, starting at line 214 (“For labeling, a 1/2 mixture of alkyne-palmitate and alkyne oleate was used, which ensures with their typical preferences for sn-1 and sn-2 positions [29], respectively, an efficient labeling of most phospholipids. Labeling in TAG is expected to distribute over all three hydroxyl groups. Under the conditions of a short labeling pulse, multiple labeling in one molecule was not observed. Note that the labeling pattern indicates the relative flow of label into the target molecules. For determination of absolute lipid synthesis rates, additional information regarding total lipid synthesis rate would be required”).

Figure 1 legend should include details about the LPS treatments and amounts administered.

Done:  If indicted (+LPS), mice were injected intraperitoneally with 15 mg/kg weight of E. coli LPS 1.5 h before liver perfusion (in green, please see the attachment)

Figure 3 legend states that hepatocytes were incubated with 16h  pre-conditioned supernatant from Kupffer cells but para 2.5 states that Kupffer cell supernatants were harvested after 2h before transfer to hepatocytes for a 16h incubation.  This apparent contradiction should be clarified.

Many thanks for the careful reading! KC supernatants were preconditioned for 16h, the hepatocytes were given 2 h to settle on the plate before the medium change. We corrected it in the Methods section.

Reviewer 3 Report

Introduction:

  1. Line 48: ‘causes of…’
  2. Line 48: Please end this sentence with a dot if you want to use the capital letter after the colon mark (‘:’).
  3. Line 64: ‘strategies’ instead of ‘challenges’
  4. Lines 69 – 70: Please move this sentence to another place of manuscript (Materials & methods or Discussion), because here is little confusing.

Materials and Methods:

  1. Lines 82 - 83: Please give an exact number of Local Ethical Committee approval of this animal experiment.
  2. Please divide this section in the following way: first you should describe an in vivo experiment (2.1 Mouse experiment) and all of the methods connected with this experiment (2.1.1 Glucose tolerance test, 2.1.2 Isolation of hepatocytes… etc.) and next in vitro experiment and all methods used in this part.
  3. It may be very interesting to analyze also free fatty acids profile of liver tissue or hepatic microsomes by chromatographic techniques (e.g. GC-MS). Please consider this in Your further research as it may provide some valuable information to explain metabolic mechanism of described dependence in NAFLD.

Author Response

Introduction:

  1. Line 48: ‘causes of…’

Done

  1. Line 48: Please end this sentence with a dot if you want to use the capital letter after the colon mark (‘:’).

Done

  1. Line 64: ‘strategies’ instead of ‘challenges’

Done

  1. Lines 69 – 70: Please move this sentence to another place of manuscript (Materials & methods or Discussion), because here is little confusing.

We removed the sentence, since it is just a one sentence summary of the papers content.

Materials and Methods:

  1. Lines 82 - 83: Please give an exact number of Local Ethical Committee approval of this animal experiment.

Done: permission LANUV NRW, 84-02.04.2015.A381

  1. Please divide this section in the following way: first you should describe an in vivo experiment (2.1 Mouse experiment) and all of the methods connected with this experiment (2.1.1 Glucose tolerance test, 2.1.2 Isolation of hepatocytes… etc.) and next in vitro experiment and all methods used in this part.

Done. We separated experiments that start with a living animal 2.1.X. from those that start with cells, 2.2.X. Statistics becomes 2.3.

  1. It may be very interesting to analyze also free fatty acids profile of liver tissue or hepatic microsomes by chromatographic techniques (e.g. GC-MS). Please consider this in Your further research as it may provide some valuable information to explain metabolic mechanism of described dependence in NAFLD.

Many thanks for this valuable suggestion. We also would like to learn more about intracellular free FA, but this is a real experimental challenge. We have an ESI-MS in the meantime but not the required GC-MS. Our major concern is to separate the high free FA content of the blood from the low intracellular free FA and to control for post-mortem release of free FA by lipases.
